# Improving robustness of spontaneous speech synthesis with linguistic speech regularization and pseudo-filled-pause insertion

*Yuta Matsunaga, Takaaki Saeki, Shinnosuke Takamichi, and Hiroshi Saruwatari*

Graduate School of Information Science and Technology, The University of Tokyo, Japan.

matsunaga-yuta339@g.ecc.u-tokyo.ac.jp,
{takaaki_saeki,shinnosuke_takamichi,hiroshi_saruwatari}@ipc.i.u-tokyo.ac.jp

## Abstract

We present a training method with linguistic speech regularization that improves the robustness of spontaneous speech synthesis methods with filled pause (FP) insertion. Spontaneous speech synthesis is aimed at producing speech with human-like disfluencies, such as FPs. Because modeling the complex data distribution of spontaneous speech with a rich FP vocabulary is challenging, the quality of FP-inserted synthetic speech is often limited. To address this issue, we present a method for synthesizing spontaneous speech that improves robustness to diverse FP insertions. Regularization is used to stabilize the synthesis of the linguistic speech (i.e., non-FP) elements. To further improve robustness to diverse FP insertions, it utilizes pseudo-FPs sampled using an FP word prediction model as well as ground-truth FPs. Our experiments demonstrated that the proposed method improves the naturalness of synthetic speech with ground-truth and predicted FPs by 0.24 and 0.26, respectively.

**Index Terms**: spontaneous speech synthesis, filled pause, spontaneity, linguistic speech regularization

## 1. Introduction

Speech synthesis is a technique for artificially synthesizing human-like speech. Rapid advances in text-to-speech (TTS) technology using deep learning have enabled the development of reading-style speech-synthesis techniques that can synthesize natural, almost human-like speech [1–4]. In contrast, spontaneous speech synthesis has proven to be a more challenging task [5–9]. Spontaneous speech has a unique characteristic, speech disfluency [10]. Spontaneous speech synthesis incorporating speech disfluency enables the synthesis of human-like speech beyond simply text reading. Our aim is to achieve speech synthesis with human-like "spontaneity" through synthesizing speech disfluency. Our particular focus is filled pauses (FPs), which are a kind of speech disfluency that play an important role in human speech [10–15].

While high-quality reading-style speech synthesis has been achieved [1–4], high-quality spontaneous speech synthesis with FPs is more difficult to achieve. This is because modeling spontaneous speech with FPs is difficult due to the complexity of data distribution such as the presence of diverse FPs. Several studies have addressed this challenge by using a speech recognition corpus [16] and restricting the FP vocabulary [9]; however, large amounts of data are required. Therefore, our goal is to achieve a spontaneous speech synthesis method that improves robustness to diverse FP insertion without scaling up the required data amount. However, the modeling of FP-included speech is difficult, and the FP-included synthesis is unstable because the FPs inserted during training are diverse and sparse. This reduces the robustness of the linguistic speech elements

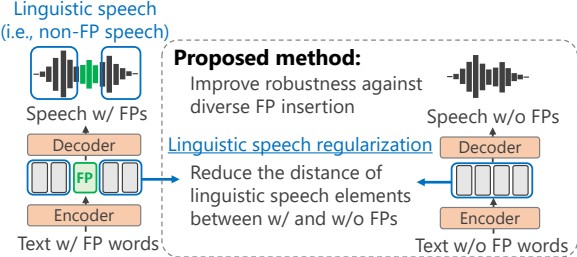

Figure 1: *Concept of proposed training method to improve robustness of spontaneous speech synthesis method against diverse FP insertions.*

(i.e., speech elements that are not FPs) against FP insertion during inference, resulting in degraded quality of the FP-included synthesized speech.

We have developed a training method with linguistic speech regularization that improves the robustness of a spontaneous speech synthesis method against diverse FP insertion (hereinafter referred to as "linguistic speech regularization"). Figure 1 shows the concept of linguistic speech regularization. It suppresses variations in linguistic speech elements between synthesized speech with and without FPs and stabilizes the synthesis of linguistic speech elements. Robustness against insertion of various FPs is further improved by performing regularization using pseudo-FPs sampled from a pre-trained prediction model as well as ground-truth FPs. We first analyzed the FP insertion impact on each module of the TTS model in the spontaneous speech synthesis method. We then subjectively evaluated our regularization method in several settings and investigated how it improves the naturalness of synthesized speech with FPs. We also evaluated how well the proposed method mitigates the FP-insertion impact for the synthesis model. Audio samples are available from our project page[1]. The key contributions of this work are as follows:

- To improve the robustness of FP-included speech synthesis to FP insertion, we developed a training method that uses regularization to stabilize the synthesis of linguistic speech elements. To further improve robustness, we use pseudo-FPs sampled using a pre-trained FP word prediction model as well as ground-truth ones.

- Our experimental results demonstrate that the proposed method improves the naturalness of synthetic speech with ground-truth and predicted FPs by approximately 0.24 and

---

[1] https://sites.google.com/g.ecc.u-tokyo.ac.jp/yuta-matsunaga/publications/fp_regularization

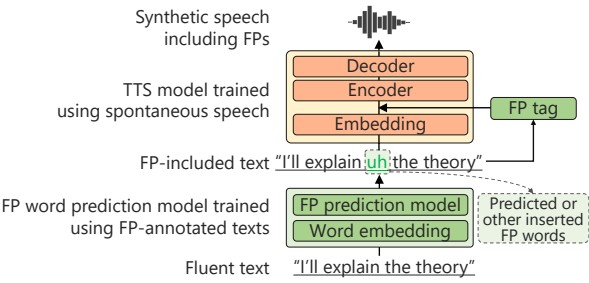

Figure 2: *Spontaneous speech synthesis method using FP word prediction model and TTS model [17].*

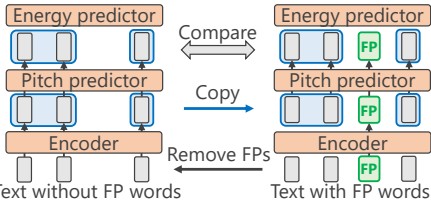

Figure 3: *Method for investigating impact of FP insertion. This example targets pitch predictor.*

0.26, respectively.

- Our analysis of the FP insertion effect for both the conventional and proposed spontaneous speech synthesis method demonstrated that the proposed method can suppress variations in linguistic speech elements with inserted FPs.

## 2. Spontaneous speech synthesis with filled pause insertion and related challenges

### 2.1. Spontaneous speech synthesis with filled pause insertion

Figure 2 illustrates the spontaneous speech synthesis method in our previous work [17]. It uses an FP word prediction model and a TTS model and synthesizes speech with FPs from text without FP words. The FP word prediction model inserts one of 13 FP words or "None" (i.e., an FP word is not inserted) into the morphologically segmented text. The TTS model then synthesizes FP-included speech from the FP word-inserted text. Both models are pre-trained on an FP-annotated spontaneous speech corpus. This method can also synthesize speech from text with other inserted FP words (e.g., ground-truth ones, which are actually used by speakers) instead of predicted ones. Fast-Speech 2 [2] was used as the TTS model. It mainly consists of an encoder, duration/pitch/energy predictors, and a decoder.

### 2.2. Related challenges

With the previous method (described in Section 2.1), FP insertion degrades the naturalness of the synthesized speech. Previous studies on neural TTS models have attempted to improve naturalness by simplifying the data distribution through training with conditional input [2, 18] and by improving the modeling method through adversarial training [4, 19, 20]. Although these studies have led to high-quality reading-style speech synthesis, high-quality spontaneous speech synthesis remains a challenge due to the complexity of the data distribution and the difficulty of modeling spontaneous speech. Chen et al. addressed these problems by using a speech recognition corpus [16], and Yan et al. substantially limited the FP word vocabulary [9]; however, these approaches require large amounts of data. Fernandez et al. also tackled these challenges by adopting a data augmentation method based on voice conversion [21]; however, this approach requires clean recordings by professional voice under controlled conditions. We aim to achieve a spontaneous speech synthesis method that is robust to a diverse FP insertion without the need for a large amount of data. However, there remains a challenge that the FPs inserted during training are diverse and sparse because of the large vocabulary and their infrequent oc-

currence compared with the linguistic content. This reduces the robustness of the linguistic speech elements to FP insertion during inference, which results in degradation of the FP-included synthesized speech.

To improve the robustness of the spontaneous speech synthesis method against FP insertion, we propose using regularization to stabilize the synthesis of the linguistic speech elements. We utilize pseudo-FPs predicted using a pre-trained FP word prediction model as well as ground-truth FPs to further improve robustness to diverse FP insertions.

## 3. Analysis method of filled pause insertion impact in synthesis model

To investigate how each module of the TTS model, described in Section 2, is affected by FP insertion, we compared intermediate representations of linguistic speech elements with and without FPs. We analyzed the intermediate representations output from each of the five modules in FastSpeech 2. The changes in representations accumulated from the first (i.e., encoder) to the last (i.e., decoder) module. We thus needed to isolate each module's effect to analyze each module separately. We did this by replacing the intermediate representations inferred from FP word-included sentences input to the targeted module with representations inferred from FP word-removed sentences. This procedure ensured that only the FP elements of the targeted module's inputs differed between with and without FPs. This enabled us to investigate the impact of only the target module by comparing the target module's outputs. Figure 3 shows an example of the investigation process for the pitch predictor. The other modules can be investigated in the same manner. Since we obtained similar results for both speakers in JLecSponSpeech, we present the results for only one speaker.

## 4. Training method with linguistic speech regularization

We propose a method using regularization for training the TTS model in the spontaneous speech synthesis method that improves model robustness to diverse FP insertion. As described in Section 2, the spontaneous speech synthesis method has low robustness to FP insertion due to the sparsity and diversity of FP appearances. To address this problem, we introduce a regularization term: the distance between intermediate representations of linguistic speech elements of synthetic speech with and without FPs. This reduces the differences between the representations predicted from text with and without FPs. To stabilize the training of a generative adversarial network [22], feature matching [23] has been proposed to reduce the differences between the intermediate representations of the discriminator predicted from real and generated data. Our regularization method,

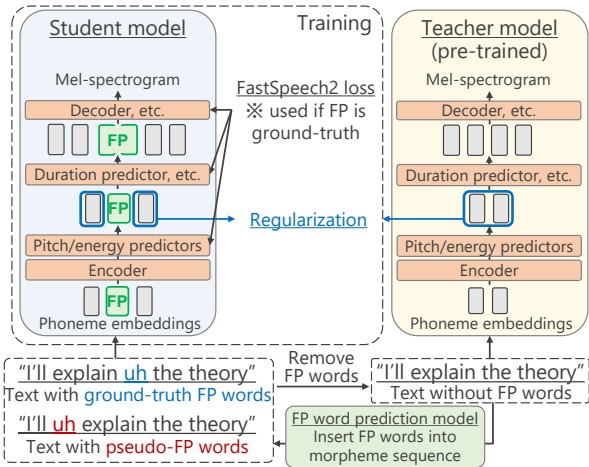

Figure 4: *Proposed training method with linguistic speech regularization for spontaneous speech synthesis method.*

which was inspired by this feature matching, suppresses variations between intermediate representations of linguistic speech elements of synthetic speech with and without FPs. It stabilizes the synthesis of the linguistic speech and thereby enables the TTS model to synthesize FP-included speech with natural linguistic speech without being greatly affected by FP insertions.

Figure 4 shows an overview of the proposed training method. Linguistic speech regularization is applied to synthesized speech with ground-truth FPs to suppress variations in the linguistic speech elements between synthesized speech with and without ground-truth FPs. It was extended to pseudo-FPs that are predicted to be inserted with high probability.

### 4.1. Basic linguistic speech regularization

First, we introduce linguistic speech regularization for ground-truth FPs. We pre-train a TTS model using FP-included spontaneous speech data as a teacher model. We then train a student model using the teacher model with fixed model parameters. In the training process, the teacher model synthesizes speech without FPs from the FP-removed phoneme inputs. The student model synthesizes speech with ground-truth FPs from ground-truth-FP-included phoneme inputs. Next, we calculate the regularization term, $R^{(\mathrm{GT})}$, which is the distance between intermediate representations of linguistic speech elements of the synthetic speech without FPs and with ground-truth FPs (using the teacher model and student model, respectively). We arbitrarily decide which intermediate representation is used for regularization (i.e., the output of the energy predictor, the pitch predictor, etc.). The loss function is written as

$$L = L^{(\mathrm{TTS})} + \alpha R^{(\mathrm{GT})}, \qquad (1)$$

$$R^{(\mathrm{GT})} = \sum_{i \in M} k_i ||\hat{\boldsymbol{h}}_i^{(\mathrm{GT})} - \hat{\boldsymbol{h}}_i^{(\mathrm{NO})}||_1, \qquad (2)$$

where $L^{(\mathrm{TTS})}$ is the loss used in the TTS model training (FastSpeech 2 here), and $\alpha$ is a hyperparameter used to control the weight of the regularization term. $\hat{\boldsymbol{h}}_i^{(\mathrm{GT})}$ and $\hat{\boldsymbol{h}}_i^{(\mathrm{NO})}$ are the linguistic speech elements of intermediate representations output from the module $i$ of the TTS model. These are inferred from text with and without ground-truth FPs, respectively. $M$

is the set of the modules in the TTS model, FastSpeech 2 [2], consisting of an encoder, duration/pitch/energy predictors, and a decoder; $k_i$ is a hyperparameter used to control the weight of the regularization for each module's outputs.

### 4.2. Linguistic speech regularization with pseudo filled pauses

We next extend the linguistic speech regularization for ground-truth FPs to that for both ground-truth and pseudo-FPs to further improve the robustness to diverse FP insertions. We obtain a text with pseudo-FP words by inserting sampled FP words (including no insertion) into FP word-removed text. These FP words are sampled on the basis of the predicted probabilities using the FP word prediction model described in Section 2.1. Note that, while the prediction model inserts the most probable FP word in inference, it inserts probabilistically sampled FP words with the regularization method. The student model then synthesizes speech with pseudo-FPs. We calculate the regularization term, $R^{(\mathrm{Pseudo})}$, which is the distance between the intermediate representations of the linguistic speech elements of the synthesized speech without FPs and with pseudo-FPs (using the teacher model and student model, respectively). The targeted intermediate representations are the same as those in Section 4.1. The loss function is written as

$$L = L^{(\mathrm{TTS})} + \alpha(R^{(\mathrm{GT})} + \beta R^{(\mathrm{Pseudo})}), \qquad (3)$$

$$R^{(\mathrm{Pseudo})} = \sum_{i \in M} l_i ||\hat{\boldsymbol{h}}_i^{(\mathrm{Pseudo})} - \hat{\boldsymbol{h}}_i^{(\mathrm{NO})}||_1, \qquad (4)$$

where $\beta$ is a hyperparameter used to control the ratio of the regularization term for pseudo-FPs to that for ground-truth ones, $\hat{\boldsymbol{h}}_i^{(\mathrm{Pseudo})}$ is the linguistic speech elements of intermediate representations output from the module $i$ of the TTS model (inferred from text with pseudo FPs), and $l_i$ is a hyperparameter used to control the weight of regularization for each module.

We also used an alternative method in which random FP words are inserted instead of probabilistically sampled ones. In this alternative method, an FP word's position in the sentence is randomly selected, and one FP word is inserted per sentence.

## 5. Experimental evaluation

### 5.1. Setting

We used the JLecSponSpeech lecture speech corpus by two Japanese speakers [17]. We downsampled the speech to 22.05 kHz and utilized 97 sentences as a test set in subjective evaluations. The other preprocessing steps, model hyperparameters, and training conditions followed those in our previous work [17].

In the analysis of the FP-insertion impact, we analyzed the conventional spontaneous speech synthesis method presented in our previous work [17]. To investigate the duration predictor, we used the normalized error, which was calculated by dividing the difference between the predicted duration of the linguistic speech elements with and without FPs by the predicted duration of the linguistic speech elements without FPs. For the encoder and decoder, we calculated the cosine similarities of the linguistic speech elements' intermediate representations between with and without FPs. For the pitch and energy predictors, we calculated the cosine similarities of the linguistic speech elements of each predictor's outputs, with the values of the residual path added. We used cosine similarity because it ranges

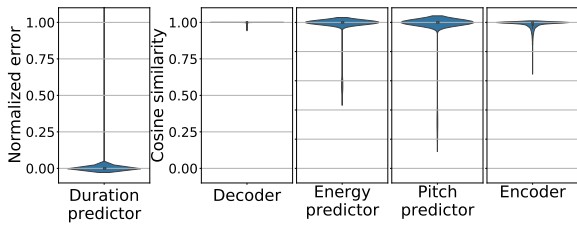

(a) **Results for five modules**

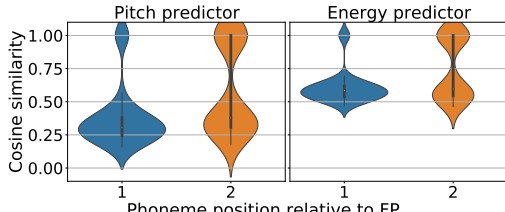

(b) **Results for pitch/energy predictor in adjacent phonemes to FPs**

Figure 5: *Results of preliminary analysis of filled pause insertion impact in spontaneous speech synthesis method. Distributions of cosine similarities or normalized errors of linguistic speech elements in outputs from each module in speech-synthesis model.*

from $-1$ to $1$ for any feature and is thus suitable for comparing how features vary among utterances. We could also use the Euclidean distance, but that would require removing the effects of the magnitude-scale variations among utterances.

In the training with the linguistic speech regularization, we used the teacher model trained under the above conditions and the student model trained using the loss function described in Section 4. We set $\alpha = 1.0$ in Equations (1) and (3) for all regularizations. The hyperparameters and other training conditions of the student model were the same as those of the teacher model. We used the "non-personalized" FP word prediction model from our previous work [24] to sample pseudo-FP words and created 128 sentences with pseudo-FP words inserted. In training, we randomly selected one of the sentences and calculated the value of the linguistic speech regularization term.

## 5.2. Preliminary analysis results

We first present the results of the analysis, described in Section 3. We analyzed the results focusing on the distribution peaks. While most frames distant from the FP frames were not affected by the FPs, the frames near the FPs were affected by the FPs and produced peaks. The mean values were considerably affected by the values of the frames distant from the FPs; thus, we focused on the peaks rather than the mean values.

As shown by the violin plots of the normalized error and cosine similarity distributions in Figure 5, the normalized errors of the duration predictor outputs and the similarities of the encoder/decoder outputs were distributed around 0.0 and 1.0, respectively. This indicates that these modules are not substantially affected by FP insertion. However, while most of the similarities of the pitch and energy predictor outputs were distributed around 1.0, we can also see peaks around 0.3 and 0.6, respectively. This indicates that these predictors especially are affected by FP insertion among the five modules. We also investigated in which phoneme positions around the FP phonemes

Table 1: *Comparison of preference scores for naturalness of speech synthesized by conventional model trained without linguistic speech regularization and by proposed one with regularization only for ground-truth FPs that targets pitch and energy predictor outputs*

| Target A | Score
Target A vs. Target B | p-value | Target B |
|---|---|---|---|
| – | 0.450 vs. **0.550** | $1.43 \times 10^{-2}$ | energy |
| – | **0.547** vs. 0.453 | $2.22 \times 10^{-2}$ | pitch and energy |
| energy | 0.523 vs. 0.477 | $2.54 \times 10^{-1}$ | pitch and energy |

the output values were affected by FP insertion for the pitch and energy predictors. As shown in Figure 5(b), the similarities are mostly distributed around 0.3 and 0.6 for the next phonemes 1 or 2 away from the FP phonemes. These results indicate that the representations after the pitch and energy predictors are affected by FP insertion in the phonemes around the FPs.

## 5.3. Evaluation of training method with linguistic speech regularization

### 5.3.1. Evaluation of basic regularization method

We first investigated the effect of the linguistic speech regularization for ground-truth FPs (Section 4.1). We conducted an AB preference test to compare the naturalness of synthesized speech samples created by the conventional and proposed models. The models were trained without and with linguistic speech regularization, respectively. We evaluated FP-included speech synthesized from the ground-truth-FP-included text (TrueFP). Each of 30 listeners evaluated 10 pairs of speech samples randomly selected from the test set described in Section 5.1. These numbers were the same in the subsequent AB preference tests.

We evaluated two methods on the basis of the results presented in Section 5.2: one targeted energy only, and the other targeted both pitch and energy. The former indirectly affected the pitch predictor, which is the module immediately before the energy predictor (i.e., the module in which the gradient in particular is back-propagated). The latter directly affected both predictors. We did not include the results of a method that only targets the pitch predictor as it had no effect on the energy predictor. We set $k_i$ in Equation (2) to 1.0 for the energy predictor and 0.0 for the other modules in the former method and set $k_i$ to 1.0 for both the pitch and energy predictors and 0.0 for the other modules in the latter method.

As shown in Table 1, while the proposed method had a significantly higher score for energy than the conventional one, the method for both pitch and energy had a significantly lower score. These results indicate that, while regularization for energy improves the naturalness of synthetic speech with FPs, that for both pitch and energy reduces the naturalness. We thus used linguistic speech regularization for only the energy predictor outputs in the subsequent experiments.

### 5.3.2. Evaluation of regularization method using pseudo-filled-pause insertion

**Evaluation of sampling methods.** We investigated the effect of using an FP word prediction model for sampling pseudo-FP words in the regularization method. We compared a model trained with regularization for pseudo-FPs sampled on the basis of the probabilities predicted by the FP word prediction model

Table 2: *Comparison of preference scores for naturalness between speech synthesized with regularization of pseudo-FPs sampled randomly and sampled probabilistically with an FP prediction model*

| Pseudo FP | Score | p-value | Pseudo FP |
|---|---|---|---|
| Random | 0.433 vs. **0.567** | $1.06 \times 10^{-3}$ | Probabilistic |

Table 3: *Comparison of preference scores for naturalness of speech synthesized by spontaneous speech synthesis models between model using regularization only for ground-truth FPs and one using regularization for both ground-truth FPs and pseudo-FPs sampled with an FP prediction model*

| $\beta$ | Score | p-value | $\beta$ |
|---|---|---|---|
| 0.0 | 0.423 vs. **0.577** | $1.63 \times 10^{-4}$ | 4.0 |

Table 4: *Evaluation results for naturalness of speech samples synthesized using spontaneous speech synthesis models: conventional model trained without regularization ("Conv.") and proposed model trained with regularization for probabilistically sampled FPs with $\alpha = 1.0$ and $\beta = 4.0$ ("Prop.")*

(a) **Preference scores from AB tests**

| Sample | Score Conv. vs. Prop. | *p*-value |
|---|---|---|
| NoFP | 0.497 vs. 0.503 | $8.71 \times 10^{-1}$ |
| TrueFP | 0.483 vs. 0.517 | $4.15 \times 10^{-1}$ |
| FP speech in TrueFP | 0.489 vs. 0.511 | $3.46 \times 10^{-1}$ |
| Linguistic speech in TrueFP | 0.407 vs. **0.593** | $4.16 \times 10^{-6}$ |
| PredFP | 0.410 vs. **0.590** | $9.16 \times 10^{-6}$ |
| FP speech in PredFP | **0.547** vs. 0.453 | $7.32 \times 10^{-5}$ |
| Linguistic speech in PredFP | 0.457 vs. **0.543** | $3.38 \times 10^{-2}$ |

(b) **Mean opinion scores. Bold score is higher one for each type of sample.**

| Method | Sample | Mean ± 95% conf. interval |
|---|---|---|
| Conv. | TrueFP | 3.350 ± 0.125 |
| | PredFP | 3.280 ± 0.125 |
| Prop. | TrueFP | **3.590 ± 0.121** |
| | PredFP | **3.537 ± 0.117** |

with a model trained with regularization for pseudo-FPs sampled randomly. We conducted an AB preference test to evaluate ground-truth-FP-included synthesized speech using those models. We set $\beta = 4.0$, the value subjectively evaluated to be the best in a preliminary experiment. As shown in Table 2, the score was significantly higher when predicted probabilities were used, demonstrating the effectiveness of using an FP word prediction model in the regularization with pseudo-FP insertion.

**Evaluation of linguistic speech regularization for pseudo-FPs.** We compared the best model trained with linguistic speech regularization for both ground-truth and pseudo-FPs (i.e., trained with $\beta = 4.0$) with a model trained with linguistic speech regularization only for ground-truth FPs (i.e., $\beta = 0.0$). We conducted an AB test to evaluate the naturalness of the FP-included speech they synthesized. As shown in Table 3, synthetic speech with regularization using both ground-truth and pseudo-FPs was evaluated as significantly higher. This indicates that synthetic speech with FPs is more natural when using regularization for both ground-truth and pseudo-FPs than when using regularization for only ground-truth FPs.

**Evaluation of conventional and proposed methods.** Finally, we evaluated a model trained using the conventional method with the most highly evaluated model trained using the proposed method. For the conventional method, the model was trained without the regularization, as in our previous study [17]. In preliminary experiments, the proposed model with $\beta = 4.0$ was evaluated as the best; thus, we used a model trained with linguistic speech regularization with $\alpha = 1.0$ and $\beta = 4.0$ as the proposed model to be evaluated.

We first conducted a comparative evaluation using AB preference tests. To investigate how the proposed method improves the naturalness of synthesized speech with FPs, we evaluated seven kinds of speech samples: speech without FPs synthesized from text without FP words (NoFP), speech with FPs synthesized from text with ground-truth FP words (TrueFP), the FP element of speech with ground-truth FPs (FP speech in TrueFP), the linguistic speech elements of ground-truth-FP-included speech (linguistic speech in TrueFP), speech with predicted FPs synthesized from text with predicted FP words (PredFP), the FP element of speech with predicted FPs (FP speech in PredFP), and the linguistic speech elements of predicted-FP-included speech (linguistic speech in PredFP). We extracted the FP and linguistic speech from the synthetic speech

with FPs by using information on the duration of each phoneme predicted by the TTS model. During inference, both the conventional and proposed models can use text with any FP word, both ground-truth and predicted ones, as inputs. We thus synthesized the speech samples using both the conventional and proposed models and compared them.

As shown in Table 4(a), the proposed method improved the naturalness of the linguistic speech elements of speech with both ground-truth and predicted FPs. In synthetic speech with predicted FPs, while the naturalness of both the linguistic speech elements and the entire speech was improved, that of the FP speech was degraded. While the naturalness of the entire synthetic speech with predicted FPs was significantly improved, that with ground-truth FPs was not.

Next, we conducted an absolute evaluation by using a five-point mean opinion score (MOS) test. Since this evaluation was aimed at investigating the naturalness of FP-containing synthetic speech, we evaluated only speech with ground-truth and predicted FPs. Each of 100 listeners evaluated 12 speech samples randomly selected from the test set. As shown in Table 4(b), compared with the conventional model, the proposed model improved the naturalness of both synthetic speech with ground-truth FPs and with predicted FPs by 0.24 and 0.26, respectively.

### 5.3.3. Discussion

We first discuss the results of regularization targeting not only the energy predictor but also the pitch predictor. As described in Section 5.3.1, performance was degraded by regularization targeting the pitch predictor as well as the energy one. Regularization applied only to the energy predictor's output also affects the pitch predictor, which is an immediately former module of the energy one, via back-propagation. Moreover, applying regularization to both the pitch and energy predictors additionally affects the pitch predictor. This results in an excessive impact of regularization on the pitch predictor because our regularization

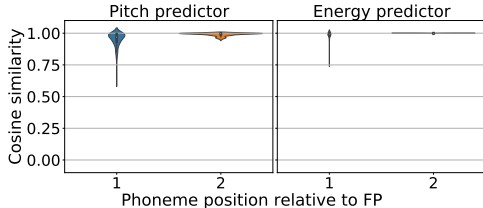

Figure 6: *Results of analysis for pitch/energy predictors. Distributions of cosine similarities of pitch/energy predictor outputs in phonemes adjacent to FPs with proposed spontaneous speech synthesis method.*

Table 5: *Mean opinion scores of synthetic speech with and without FPs for best proposed model (i.e., trained with consistency loss for probabilistically sampled FPs with $\alpha = 1.0$ and $\beta = 4.0$). Bold score is higher one for each speaker*

| Criterion | Utterance | Mean ± 95% conf. interval | |
| --- | --- | --- | --- |
| | | Spk. A | Spk. B |
| Naturalness | NoFP | **3.470 ± 0.110** | **3.587 ± 0.100** |
| | TrueFP | 3.433 ± 0.108 | 3.520 ± 0.106 |
| Spontaneity | NoFP | 3.267 ± 0.121 | 3.283 ± 0.123 |
| | TrueFP | **3.367 ± 0.123** | **3.397 ± 0.116** |

method is aimed at mitigating the effect of FP insertion, rather than completely eliminating it.

The results in Table 4(a) show that naturalness improved with PredFP, whereas there was no significant difference with TrueFP. This suggests that the proposed method focuses on improving the naturalness of speech with predicted FPs rather than speech with ground-truth FPs. Moreover, the FP elements were degraded for PredFP. This might be because the regularization targets only the linguistic speech elements and thus cannot train the predicted FP elements. Future work will focus on improving the naturalness of FP speech by data augmentation.

### 5.4. Anaylsis results for proposed method's effectiveness

We analyzed how the proposed method mitigated the FP-insertion impact for the pitch and energy predictors (i.e., the modules most affected by FP insertion). We investigated in which phoneme positions around the FP phonemes the output values were affected by FP insertion, in the same manner as described in Section 3. We then compared the results between the conventional and proposed model. Figure 6 shows the results for the proposed model. The results for the conventional model (Section 5.2) indicate that the representations after the pitch and energy predictors are affected by FP insertion in the phonemes around the FPs. On the other hand, with the proposed model, the similarities are mostly distributed around 1.0, indicating that the intermediate representations of FP-included synthesis are not affected by FP insertion. This demonstrates that the proposed method can suppress variations in linguistic speech elements with FP insertion.

### 5.5. Evaluation of filled pause effects on synthetic speech

We investigated the effectiveness of using FPs in speech synthesis. Since an FP is a characteristic of spontaneous speech, we evaluated the effect of FP presence on the perception of "spontaneity" as well as naturalness in synthetic speech. We used the criteria used in the previous study [25] for spontaneity. We conducted an absolute evaluation by using a five-point MOS test to evaluate the synthesized speech of each speaker in JLecSpon-Speech. Each of 100 listeners evaluated 6 speech samples. As shown in Table 5, the speech without FPs had higher scores for naturalness than that with ground-truth FPs. This might be because the proposed method focuses on improving linguistic speech elements, and the naturalness of FP speech is still low. On the other hand, speech with ground-truth FPs had higher scores for spontaneity than that without FPs, suggesting that speech with FPs is perceived as more spontaneous. These results are consistent with those of a previous study [26], which found that disfluent speech is perceived as more spontaneous than fluent speech. Considering the results for naturalness, further improvement in the quality of the FP speech might improve the perception of spontaneity for FP-included speech.

## 6. Conclusion

Our proposed training method with linguistic speech regularization improves the robustness of the spontaneous speech synthesis method. Our experimental results demonstrated that the proposed method improves the naturalness of the entire synthetic speech with predicted FPs as well as the linguistic speech elements of synthetic speech with ground-truth and predicted FPs. However, the FP elements of the synthesized speech were not improved. Future work will focus on improving the FP speech quality by data augmentation of FP-included speech.

## 7. Acknowledgements

This work was supported by JST, Moonshot R&D Grant Number JPMJMS2011.

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
