# OpenReview forum: "Improving robustness of spontaneous speech synthesis with linguistic speech regularization and pseudo-filled-pause insertion"
_Interspeech.org/2023/Workshop/SSW — SSW12_

### Official Review · Reviewer_AmfT · 2023-06-01
**Inspired by work in feature-matching regularization, this paper proposes a very similar regularization approach to alleviate the instability observed when synthesizing filled pauses (learned from a modest amount of training data).**

**Rating:** 5
**Confidence:** 5

**Review:**

Inspired by work in feature-matching regularization, this paper proposes a very similar regularization approach to alleviate the instability observed when synthesizing filled pauses (learned from a modest amount of training data). The key idea is to match the learned internal embeddings between two systems: one that has not been exposed to fillers, and one that has, using parallel training data (with and without fillers).  The topic is an important one, as naturally synthesizing spontaneous speech and imitating some of the discourse mechanisms that humans deploy in conversation (such as filled pauses) remain challenging for TTS.

The key idea is summarized by the authors as follows: “It suppresses variations in linguistic speech elements between synthesized speech with and without FPs and stabilizes the synthesis of linguistic speech elements.” – My main issue with the fundamental idea of this proposal is: is it at all well justified? The insertion of an FP can alter the phrasal structure of a sentence, the corresponding intonation, and the duration of the phones in the vicinity of an FP. It seems that imposing a matching loss that encourages the hidden representations over the non-FP region to be the same is not theoretically justified a priori (at least in the case where the loss is evaluated over the full non-FP region – theoretically, it might make more sense to at least weigh that loss dynamically as a function of how far from the insertion point you are). In fact, Table 1 confirms that regularizing the pitch representation hurts the performance (which is consistent with my observation), and only energy regularization seems to help at all. All subsequent experiments in the paper only make use of this type of regularization, which dilutes the claims of the paper (which ends up being about energy normalization only).

There are some issues with evaluation methodology:
- The use of forced-choice, binary preference tests that do not offer a “neutral/don’t-care” choice. Considering how close the results are in many of these comparisons, that would’ve been more instructive.
- Very small sample size: 12 speech samples for MOS tests; 10 pairs in Section 3.1
- “We extracted the FP and linguistic speech from the synthetic speech with FPs by using information on the duration of each phoneme predicted by the TTS model” – I’m not sure if I understand correctly, but does this mean that known timing information was used to “edit” portions of the output waveform, so that listeners could evaluate only portions of them? If this is indeed the case, why should you expect this to be an artifact-free method and not to impact the results? Related: I also found the description/definitions of the various systems in Table 4 (described in 5.3.2 – Evaluation of best proposed method) extremely confusing.

In terms of writing, the paper contains a tremendous amount of redundancy, and could be served from careful editing to streamline the presentation. Points previously discussed are brought up again in later sections, sometimes using very similar language. For instance, the majority of the discussion in 2.2 seemed to reiterate previous content, and I couldn’t tell if 5.3.3 was adding any content that hadn’t already been discussed.

Other questions/comments
- Why is the analysis of section 5.2 confined to phonemes after the insertion and not before?
- Consider relabeling the information on Table 1 in a different way. I found the two columns labeled Target initially rather confusing

A couple of relevant missing references that address the synthesis of fillers:
- “Adaptive Text to Speech for Spontaneous Style” Yan et al., Interspeech 2021, pp. 4668-4672
- “Transplantation of Conversational Speaking Style with Interjections in Sequence-to-Sequence Speech Synthesis” Fernandez et al., Interspeech 2022, pp. 5488-5492

Minor edits:
- proven to be more challenging task --> proven to be a more challenging task
- as is had no effect --> as it had no effect
- for ground-truth FPs to that for both ground-truth and pseudo-FPs to further improve --> for ground-truth FPs also to pseudo-FPs to further improve
- A couple of capitalization issues in the bibliography: gans --> GANs / tts --> TTS / lm --> LM

---

### Official Review · Reviewer_QG9V · 2023-06-05
**New method for modelling filled pauses with extensive evaluation**

**Rating:** 7
**Confidence:** 4

**Review:**

quality: The paper contains an extended evaluation of the proposed methods, bs first focusing on the five modules and identifying pitch/energy as possible problems affected by FP insertion. Next the FP sampling methods are evaluated. Finally the best method is evaluated and it's effectivenes evaluated.

clarity: The paper is sometimes hard to read because the text is not structured or things are not formalized, e.g.
in "Evaluation of best proposed method" there are a lot of different criteria presented in running text. It contains some
redundancies, repeating information.

originality: The paper proposes a regularization method for improving synthesis of filled pauses (FP). The method still
has problems with the quality of filled pauses although the quality of "linguistic speech" improves. Regularization is performed on the 5 different modules of Fastspeech 2.

significance: The task of modeling FPs with little amount of data is significant and challenging.

minor changes:

In Figure 1: The location of the phrase "Linguistic speech (i.e. non-FP speech) is mismelading.

p1 "to be more" -> "to be a more"

---

### Decision · Program_Chairs · 2023-06-14

**Decision:**

Accept

**Comment:**

SSW2003 received 45 papers. The acceptance rate is 82%. We are pleased to inform you that your paper has been accepted by the SSW2023 Program Committee. Please read the reviews carefully and submit your camera-ready paper by June 28th. Most reviewers performed a detailed review. Please answer to their questions and consider their comments. Note that camera-ready papers are credited with one extra page to allow authors to consider reviewers’ suggestions. So max 7 pages in total including figures & refs.
The deadline for submitting the revised version (with full non anonymized authors and refs!) is 28th June.